# Safety and immunogenicity of a tetravalent and bivalent SARS-CoV-2 protein booster vaccine in men

Suad Hannawi ®[1], Linda Saf Eldin ®[2], Alaa Abuquta ®[1], Ahmad Alamadi[3], Sally A. Mahmoud ®[4], Aala Hassan[1], Shuping Xu[5], Jian Li[5], Dongfang Liu[5], Adam Abdul Hakeem Baidoo[5], Dima Ibrahim[6], Mojtaba Alhaj[7], Yuanxin Chen[5], Qiang Zhou[5] & Liangzhi Xie ®[5,8] ✉

The safety and immunogenicity of a protein-based tetravalent vaccine SCTV01E that contains spike protein ectodomain (S-ECD) of Alpha, Beta, Delta and Omicron BA.1 are assessed and compared with bivalent protein vaccine SCTV01C (Alpha and Beta variants) and monovalent mRNA vaccine (NCT05323461). The primary endpoints are the geometric mean titers (GMT) of live virus neutralizing antibodies (nAb) to Delta (B.1.617.2) and Omicron BA.1 at day 28 post-injection. The secondary endpoints include the safety, day 180 GMTs against Delta and Omicron BA.1, day 28 GMTs to BA.5, and seroresponse rates of neutralizing antibodies and T cell responses at day 28 post-injection. 450 participants, comprising of 449 males and 1 female, with a median age (range) of 27 (18–62) years, are assigned to receive one booster dose of BNT162b2, 20 μg SCTV01C or 30 μg SCTV01E and completed 4-week follow-up. All SCTV01E related adverse events (AEs) are mild or moderate and no Grade ≥3 AE, serious AE or new safety concerns are identified. Day 28 GMT of live virus neutralizing antibodies and seroresponse against Omicron BA.1 and BA.5 with SCTV01E are significantly higher than those with SCTV01C and BNT162b2. These data indicate an overall neutralization superiority with tetravalent booster immunization in men.

More than three years after the COVID-19 pandemic began, the incessant evolution and emergence of new SARS-CoV-2 variants have held a tight grip on the world[1]. Omicron and its sublineages have emerged as the most antigenically divergent variant to date with >30 mutations in the spike protein, 15 of which are clustered in the receptor binding domain. Studies that investigated the effectiveness of primary and booster vaccination with approved vaccines have shown decreased efficacy against Omicron and its sublineages and waning immunity over time, although protection against hospitalization and severe disease are maintained[2–7].

Multivalent vaccine increases the diversity of antibody responses and may improve cross-strain protection. The WHO Technical Advisory Group on COVID-19 Vaccine Composition (TAG-CO-VAC) and the 175th meeting of the Vaccines and Related Biological Products Advisory Committee (VRBPAC) on June 28, 2022 have recommended developing multivalent or broad-protective vaccines against SARS-

[1]Internal Medicine Department, Al Kuwait-Dubai (ALBaraha) Hospital, Dubai, United Arab Emirates. [2]General Surgery Department, Al Kuwait-Dubai (ALBaraha) Hospital, Dubai, United Arab Emirates. [3]Ear, Nose and Throat Department (ENT), Al Kuwait-Dubai (ALBaraha) Hospital, Dubai, United Arab Emirates. [4]Biogenix labs, G42 Healthcare, Dubai, United Arab Emirates. [5]Beijing Engineering Research Center of Protein and Antibody, Sinocelltech Ltd., Beijing, China. [6]Infectious Diseases Department, Burjeel Medical City, Abu Dhabi, United Arab Emirates. [7]Research Department, Burjeel Medical City, Abu Dhabi, United Arab Emirates. [8]Cell Culture Engineering Center, Chinese Academy of Medical Sciences & Peking Union Medical College, Beijing, China. ✉e-mail: LX@sinocelltech.com

CoV-2 current and future variants and updating the vaccine strain compositions[8]. Moderna recently reported encouraging immunogenicity data on mRNA-1273.211(original and Beta variant), mRNA-1273.214 (original and Omicron B.1.1.529) and mRNA-1237.222 (original and Omicron BA.4/5)[9–11]. Likewise, Pfizer also reported on its bivalent mRNA vaccines (original and Omicron BA.1 or BA.4/5)[12]. Both reports showed the superiority of neutralizing antibody (nAb) against Omicron BA.1 and similar nAb status against the original strain compared to their monovalent progenitor vaccines.

We have previously reported the results of three phase 1/2 safety and immunogenicity trials of a protein-based bivalent adjuvanted vaccine SCTV01C containing equal amounts of spike protein ectodomain (S-ECD) of SARS-VoC-2 Alpha and Beta variants. SCTV01C was administered as a two-dose primary series (NCT 05148091) in vaccine naïve people and one booster dose in people previously vaccinated with the inactivated vaccine (NCT 05043285) and mRNA vaccine (NCT 05043311) demonstrated favorable safety and tolerability profiles in a total 922 participants, and induced high levels of spike-protein binding IgG and broad neutralizing antibody responses against Alpha, Beta, Delta and Omicron variants[13–15]. On December 2, 2022, SCTV01C was granted Emergency Use Authorization (EUA) by the National Health Commission of the People's Republic of China as a booster dose, and as a primary dose for individuals who have already been infected during the COVID-19 pandemic.

SCTV01E was manufactured by the same process as SCTV01C but has a tetravalent design containing a blend of Spike-ECD proteins derived from SARS-CoV-2 variants, Alpha (B.1.1.7), Beta (B.1.351), Delta (B.1.617.2), and Omicron BA.1., in a proportion of 1:1:1:3, with a total quantity of 30 µg. The selection of a 1:1:1:3 antigen ratio was based on empirical animal data indicating that a higher dose of Omicron BA.1 antigen is required to elicit an optimal immune response as a booster vaccine against the newer BA.1 variant. Both SCTV01C and SCTV01E are adjuvanted with a squalene-based oil-in-water emulsion SCT-VA02B to boost the immune responses and possess a trimerization auxiliary domain (T4-Foldon) to stabilize the trimeric protein conformation, exhibiting temperature stable at 25 °C for over six months and at 2–8 °C for over 24 months[16, 17].

Herein, we present the interim analysis results of the safety and immunogenicity of one booster dose of SCTV01E in people that had previously received authorized mRNA vaccines, using SCTV01C and the ancestral strain monovalent mRNA vaccine as controls, from an ongoing phase 3 study.

## Results

### Demographic and baseline characteristics

Between May 30, 2022 and September 28, 2022, 451 participants who had a prior diagnosis of COVID-19 and/or received BNT162b2 vaccines were enrolled and 149, 154 and 147 participants were assigned to receive one dose of BNT162b2, 20 µg SCTV01C and 30 µg SCTV01E (Fig. 1 and Supplementary Table 1), respectively (One participant withdrew before vaccination). Notably, six participants in the BNT162b2 group, four in the SCTV01C group, and eight in the SCTV01E group were excluded from the immunogenicity analysis due to missed or out-of-window scheduled visits (Fig. 1). Out of 451 participants, only four had chronic medical conditions (diabetes), with three in the BNT162b2 group and one in the SCTV01C group (Supplementary Table 2). All participants completed Day 28 visit and the median (min, max) time of follow-up was 73 (60, 79) days (Fig. 1). Notably, no cases of COVID-19 infection were reported during the available follow-up period when the data was locked for analysis. The median (min, max) age was 27 (18, 62) years old. 0.4%, 95.6%, and 4.0% of participants had previously received 1, 2 and 3 doses of mRNA vaccine respectively. 3.5% of all participants were previously diagnosed with COVID-19. The demographic and baseline characteristics were generally comparable for participants across all the groups. For all trial participants, the

interval between investigational vaccination and prior COVID-19 vaccination were 3–5 months (11.3%), 6–8 months (31.3%), 9–12 months (25.5%) and 13–24 months (31.9%), respectively. Regarding sex and gender, the study considered sex in its design and relied on self-reported information from participants to determine their sex. As only one female participant enrolled, there was insufficient data to carry out a sex or gender analysis. Generally, participants in each group had similar intervals from prior vaccination (Supplementary Table 1).

### Adverse events

For the SCTV01E group, all vaccine-related adverse events (AEs) were mild or moderate. There were no vaccine-related AEs with frequency ≥10%, Grade ≥3 AE, serious AE (SAE) and AE of special interest (AESI) reported. The overall incidence of adverse reactions was similar or numerically lower with the SCTV01E compared to those with SCTV01C and BNT162b2. In BNT162b2, SCTV01C and SCTV01E groups, 25 (16.8%), 27 (17.5%) and 18 (12.2%) participants experienced at least one treatment emerged adverse event (TEAE) and 19 (12.8%), 24 (15.6%) and 14 (9.5%) participants experienced at least one treatment-related adverse event (TRAE), respectively. The frequencies of solicited AEs were 15 (10.1%) in the BNT162b2 group, 19 (12.3%) in the SCTV01C group, and 7 (4.8%) in the SCTV01E group. The occurrences of vaccine-related unsolicited AEs within 28 days after the injection were also numerically lower in the SCTV01E group (4.8%) compared to those in SCTV01C (7.5%) and BNT162b2 groups (6.0%). For all three groups, the most frequent solicited AEs included pain at the injection-site, headache and pyrexia (Table 1 and Supplementary Fig. 1).

### GMT of live virus nAb against Omicron BA.1, BA.5 and Delta

At day 28 after vaccination, the GMT (95% CI) of live virus nAb against: Omicron BA.1 were 1049 (923, 1193) with 4.06-fold change over baseline, 1189 (1027, 1376) with 3.60-fold change over baseline and 1659 (1445, 1904) with 5.96-fold change over baseline; Omicron BA.5 were 1687 (1471, 1936) with 4.34-fold change over baseline, 1736 (1517, 1987) with 3.19-fold change over baseline and 2281 (1993, 2610) with 4.94-fold change over baseline in BNT162b2, SCTV01C, and SCTV01E groups, respectively. The GMRs (95% CI) of SCTV01E / BNT162b2 and SCTV01E / SCTV01C were: 1.55 (1.30, 1.85) and 1.44 (1.19, 1.74) against Omicron BA.1; 1.28 (1.07, 1.54) and 1.33 (1.10, 1.61) against Omicron BA.5, respectively. The pre-specified statistical success criteria were met for the superiority of GMTs with SCTV01E against Omicron BA.1 and BA.5 compared to those with BNT162b2. In addition, the post-hoc analysis indicated that the GMTs against Omicron BA.1 and BA.5 with SCTV01E were significantly higher than those with SCTV01C. SCTV01C showed non-inferior GMT against Omicron BA.1 (GMR = 1.04 (0.87, 1.24)) and BA.5 (GMR = 0.92 (0.77, 1.10)) compared to BNT162b2. GMTs against Delta variant were 3310 (2918, 3754) with 2.75-fold change over baseline, 3270 (2844, 3760) with 2.03-fold change over baseline, and 3873 (3365, 4456) with 2.77-fold change over baseline in BNT162b2, SCTV01C, and SCTV01E groups, respectively. The GMTs against Delta were non-inferior across the three groups (SCTV01E / BNT162b2 = 1.10 (0.92, 1.30); SCTV01E / SCTV01C = 1.21 (1.00, 1.46); SCTV01C / BNT162b2 = 0.88 (0.74, 1.05) (Fig. 2 and Supplementary Tables 3 and 4). The participants were stratified based on the time intervals between the previous and study vaccination, the number of prior doses of COVID-19 vaccine and the history of COVID-19. For participants who had received two prior doses of BNT162b2 and had no history of COVID-19, SCTV01E showed the superiority of neutralizing antibody responses against Omicron BA.1 and BA.5 compared to those with BNT162b2 (Supplementary Tables 5–7 and Supplementary Fig. 2–4).

### Seroresponse of nAb against Omicron BA.1, BA.5 and Delta

At day 28 post injection, the seroresponse rates of live virus nAb against variants: Omicron BA.1 were 61.5%, 52.0% and 76.3%; Omicron BA.5 were 66.4%, 46.0% and 63.0%; Delta were 45.5%, 34.7% and 48.9%

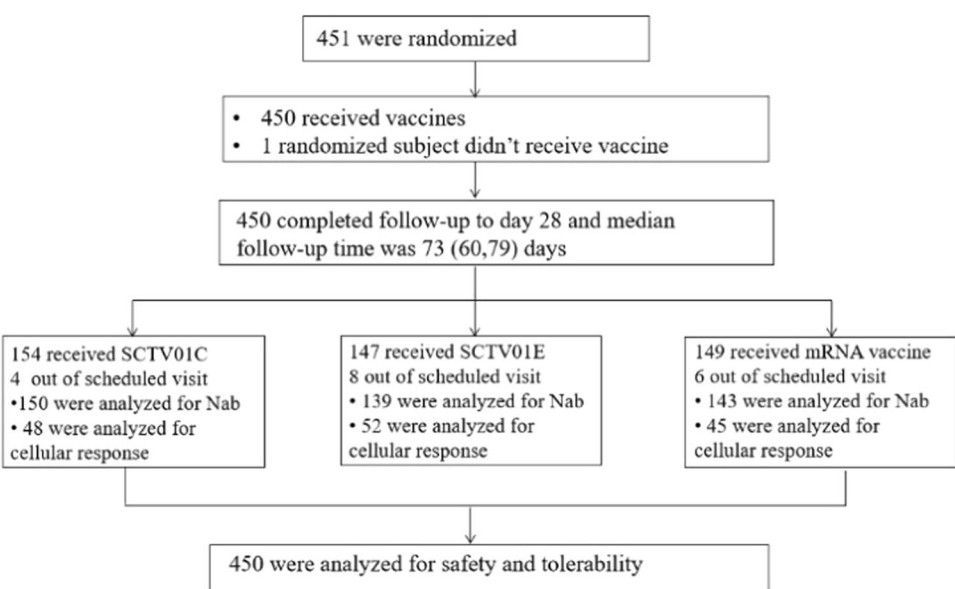

**Fig. 1 | Flow diagram of the participants.** Nab, neutralizing antibody.

in BNT162b2, SCTV01C and SCTV01E groups, respectively (Table 2). SCTV01E elicited significantly higher seroresponse rates against Omicron BA.1 than those with BNT162b2 and SCTV01C (SCTV01E minus BNT162b2 and SCTV01E minus SCTV01C, *p* <0.01).

**Post hoc analysis of nAb responses to Omicron BA.1 and BA.5**
To analyze the influence of pre-existing nAb on vaccine immunogenicity, participants were divided into three groups based on their pre-dose GMT levels for each specific variant: low baseline titer group (equal or lower than four times of the LLOQ, ≤80), medium baseline titer group (80–320), and high baseline titer group (>320) (Fig. 3). Day 28 GMTs of SCTV01E against BA.1 were: 1627, 1347 and 2560, with 28.67, 5.58 and 2.67-fold change over baseline; BA.5 were 2153, 2010 and 2765 with 28.51, 8.67 and 2.78-fold change over baseline for the low, medium and high baseline titer groups, respectively. The nAb responses with SCTV01E were consistently superior to those with SCTV01C and BNT162b2, irrespective of baseline GMTs levels of the participants. In addition, SCTV01E elicited relatively uniform GMTs across different baseline groups as compared to BNT162b2 showed a 2.55 and 2.79-fold lower in the low baseline groups than those in high baseline groups for Omicron BA.1 and BA.5 on day 28, respectively.

**T cell responses**
The peripheral blood mononuclear cells were collected to assess specific Th1 (IFN-γ release) and Th2 (IL-4 release) responses. At day 28 post injection, the mean number of IFN-γ expressing T cell increased by 1.3, 1.2, and 1.2-fold change and IL-4 expressing T cell 1.3, 1.1, and 1.4-fold change to baseline for BNT162b2, SCTV01C and SCTV01E groups, respectively (Supplementary Fig. 5).

## Discussion
A tetravalent protein-based vaccine, SCTV01E designed to provide broad-protection against SARS-CoV-2 variants is currently being evaluated in an ongoing positive-controlled phase 3 trial, using its progenitor bivalent vaccine SCTV01C and mRNA vaccine as the controls.

The tetravalent vaccine SCTV01E was developed as a modified version of the bivalent (Alpha + Beta) vaccine SCTV01C by adding two subsequent variants of concern Delta and Omicron BA.1. During this clinical study, SCTV01C demonstrated significant cross-neutralizing capability against Omicron BA.1 and BA.5 variants which emerged two years after its initial development. SCTV01E showed even greater

breadth of cross-neutralizing capabilities against a variety of Omicron variants during pre-clinical studies[18]. Seven clinical trials have been conducted for both SCTV01C and/or SCTV01E, collectively demonstrating their potential as an important vaccine platform in the context of the challenging epidemiological situation where multiple major variants are prevalent simultaneously. The flexibility of this platform enables rapid replacement of up to four new variant antigens to adapt to immune-evading variants. The findings of this investigation suggest that a tetravalent recombinant protein may be an effective approach to address both current and potential future epidemiological challenges. Currently, a phase 3 efficacy study with SCTV01E is underway in China (NCT05308576).

The results of the interim analysis indicate that the tetravalent vaccine SCTV01E given to individuals who previously received two or three doses of an authorized mRNA vaccine has a clinically acceptable safety and tolerability profile. All vaccine-related AEs were mild or moderate (Grade 1–2). There were no Grade ≥3 AE, SAE or AESI reported in the SCTV01E group. The incidence of adverse reactions was similar or numerically lower with the SCTV01E compared to those with SCTV01C and BNT162b2. These findings are consistent with previous clinical studies of SCTV01C[13–15], which identified no new safety concerns. It is important to note that during the repeat-dose toxicity test of SCTV01E in rats, certain abnormalities were observed. These included increases in neutrophilic and eosinophils, fibrinogen, and globulin, as well as decreases in reticulocyte and albumin levels. In addition, glomerulonephritis was observed in the kidneys of two out of twenty rats, however, these changes were not observed in the present trial.

In this study, 16.8% of participants in the BNT162b2 group reported experiencing at least one treatment-emergent AE, and 19 individuals (12.8%) experienced at least one treatment-related AE. The total frequency of AEs in this study is comparable to that reported in a phase 3 trial of BNT162b2 booster, which found that among 5050 participants, 25.0% experienced at least one AE after receiving a third dose of the vaccine, with 23.4% being related to vaccine administration[19]. However, these incidence rates are much lower than those reported in the Vaccines and Related Biological Products Advisory Committee Briefing Document (17 September 2021), which demonstrated that, within one month following the administration of 3rd dose of BNT162b2 vaccine, 77.2% of 306 participants reported any systemic reaction[20]. Possible reasons for these inconsistencies could

**Table 1 | Summary of AEs**

| | BNT162b2 (N = 149) n (%) | SCTV01C 20 µg (N = 154) n (%) | SCTV01E 30 µg (N = 147) n (%) |
|---|---|---|---|
| TEAEs | 25 (16.8) | 27 (17.5) | 18 (12.2) |
| TRAEs | 19 (12.8) | 24 (15.6) | 14 (9.5) |
| ≥Grade 3 AEs | 1 (0.7) | 1 (0.6) | 0 |
| ≥Grade 3 TRAEs | 0 | 1 (0.6) | 0 |
| Solicited AEs | 15 (10.1) | 19 (12.3) | 7 (4.8) |
| IP related solicited AEs | 15 (10.1) | 19 (12.3) | 7 (4.8) |
| Solicited systemic AEs | 7 (4.7) | 10 (6.5) | 6 (4.1) |
| Grade 1 | 4 (2.7) | 9 (5.8) | 5 (3.4) |
| Grade 2 | 2 (1.3) | 0 | 1 (0.7) |
| ≥Grade 3 | 1 (0.7) | 1 (0.6) | 0 |
| Headache | 3 (2.0) | 4 (2.6) | 1 (0.7) |
| Grade 1 | 2 (1.3) | 4 (2.6) | 1 (0.7) |
| Grade 2 | 1 (0.7) | 0 | 0 |
| ≥Grade 3 | 0 | 0 | 0 |
| Pyrexia | 3 (2.0) | 2 (1.3) | 3 (2.0) |
| Grade 1 | 1 (0.7) | 1 (0.6) | 2 (1.4) |
| Grade 2 | 1 (0.7) | 0 | 1 (0.7) |
| ≥Grade 3 | 1 (0.7) | 1 (0.6) | 0 |
| Myalgia | 3 (2.0) | 1 (0.6) | 2 (1.4) |
| Grade 1 | 2 (1.3) | 1 (0.6) | 2 (1.4) |
| Grade 2 | 1 (0.7) | 0 | 0 |
| ≥Grade 3 | 0 | 0 | 0 |
| Fatigue | 0 | 2 (1.3) | 1 (0.7) |
| Grade 1 | 0 | 2 (1.3) | 1 (0.7) |
| Grade 2 | 0 | 0 | 0 |
| ≥Grade 3 | 0 | 0 | 0 |
| Chills | 0 | 1 (0.6) | 0 |
| Grade 1 | 0 | 1 (0.6) | 0 |
| Grade 2 | 0 | 0 | 0 |
| ≥Grade 3 | 0 | 0 | 0 |
| Solicited local AEs | 8 (5.4) | 11 (7.1) | 1 (0.7) |
| Injection site pain | 8 (5.4) | 11 (7.1) | 1 (0.7) |
| Injection site induration | 1 (0.7) | 0 | 0 |
| Injection site swelling | 1 (0.7) | 0 | 0 |
| Unsolicited AEs | 16 (10.7) | 11 (7.1) | 11 (7.5) |
| IP related unsolicited AEs | 9 (6.0) | 7 (4.5) | 7 (4.8) |
| AESI | 0 | 0 | 0 |

*AE* adverse event, *TEAE* treatment emerged adverse event, *TRAE* treatment-related adverse event, *IP* investigational product, *AESI* adverse event of special interest.

include differences in the definition, measurement, and reporting of AEs across different studies, as well as variations in population characteristics like age distribution, comorbidities, prior vaccination, and infection history. It is worth noting that the high rate of prior infections and the predominance of young male participants in this trial might have contributed to the lower occurrence of AEs observed.

The strong correlation between the viral neutralizing antibody level and the protection from symptomatic SARS-CoV-2 has been shown in vaccinated people[21,22]. This study evaluates the live virus nAb GMTs and seroresponse rates against Omicron BA.1, BA.5 and Delta variants. At day 28 post vaccination, GMTs of nAb against: Omicron BA.1 were 4.06, 3.60, and 5.96-fold change over baseline; Omicron

BA.5 were 4.34, 3.19 and 4.94-fold change from baseline in BNT162b2, SCTV01C and SCTV01E groups, respectively. The superiority immunogenicity objectives were met for GMRs of SCTV01E / BNT162b2 and SCTV01E / SCTV01C against Omicron BA.1 and BA.5. Similarly, SCTV01E elicited significantly higher seroresponse rates for Omicron BA.1 than those with BNT162b2 and SCTV01C, based on the predefined definition for seroresponse. While statistically significant differences in post-booster antibody titers were observed between study groups, further clinical evidence is needed to demonstrate whether the numerically higher antibody titers would lead to superiority in clinical efficacy or durability of protection.

The data showed highly diversified neutralizing antibody titers to both Delta and Omicron variants at baseline. We conducted post hoc analyses to evaluate the impact of the pre-existing SARS-COV-2 immunity on the nAb responses. The participants were assigned to three groups based on their pre-dose GMTs levels. The nAb responses with SCTV01E were consistently superior to those with SCTV01C and BNT162b2, irrespective of the baseline GMTs levels of the participants. Notably, SCTV01E induced high GMTs in the participants with a low baseline that were comparable to those with high baseline titers. Having the capability to boost immune responses in persons with low baseline nAb is important, given the fact that breakthrough infections can occur in vaccinated persons. To a larger extent, individuals with low nAb levels have much higher risk of subsequent infection compared with high nAb individuals. The underlying mechanisms of enhanced nAb responses with multivalent vaccines have yet to be elucidated but could be associated with generation of immune memory and evolution of the humoral responses[23].

The study had several limitations. First, the trial was conducted in an environment of high Omicron variant circulation, and a large portion of the trial participants might have asymptomatic infection according to published reports[24,25]. Our study revealed a wide range of baseline neutralizing antibody titers against the Omicron variant. Notably, the GMT levels were considerably higher than those reported in earlier studies investigating individuals vaccinated with two doses of mRNA vaccines. However, there was no standard way to differentiate asymptomatic individuals. Second, the immunogenicity of booster vaccination was assessed in a short period, and as result, the immune persistence data is not yet available. In addition, the study was designed to evaluate the prominent circulating variants, thus, nAb responses to the ancestral SARS-CoV-2 (D614G) and vaccine prototype variants (Alpha and beta) were not assessed in this study. In addition, the study's sample population was mostly composed of young male adults. This lack of diversity may affect the generalizability and applicability of the study results. Although previous clinical studies involving SCTV01C did not reveal any significant differences in AEs or immunogenicity between male and female participants or between younger and older adults, further investigations on SCTV01E with a more balanced demographic representation are necessary.

In summary, 30 µg tetravalent protein vaccine SCTV01E, when administered to individuals who previously received mRNA vaccines had a clinically acceptable safety and tolerability profile; induced uniformly high nAb responses against Omicron BA.1, BA.5 and Delta variant, showing immunogenicity superiority to those with bivalent vaccine SCTV01C and BNT162b2. The tetravalent vaccine may be a new tool to respond to the continuous emergence of SARS-CoV-2 variants.

## Methods
### Study design and participants
This ongoing randomized, double-blinded, and positive-controlled phase 3 booster study is being conducted at Al Kuwait Hospital, Emirates Health Services in Dubai, and Burjeel Medical City in Abu Dhabi, United Arab Emirates (UAE) between May 30, 2022 and September 28, 2022. The study included two cohorts that aimed to evaluate the immunogenicity and safety of one booster dose of

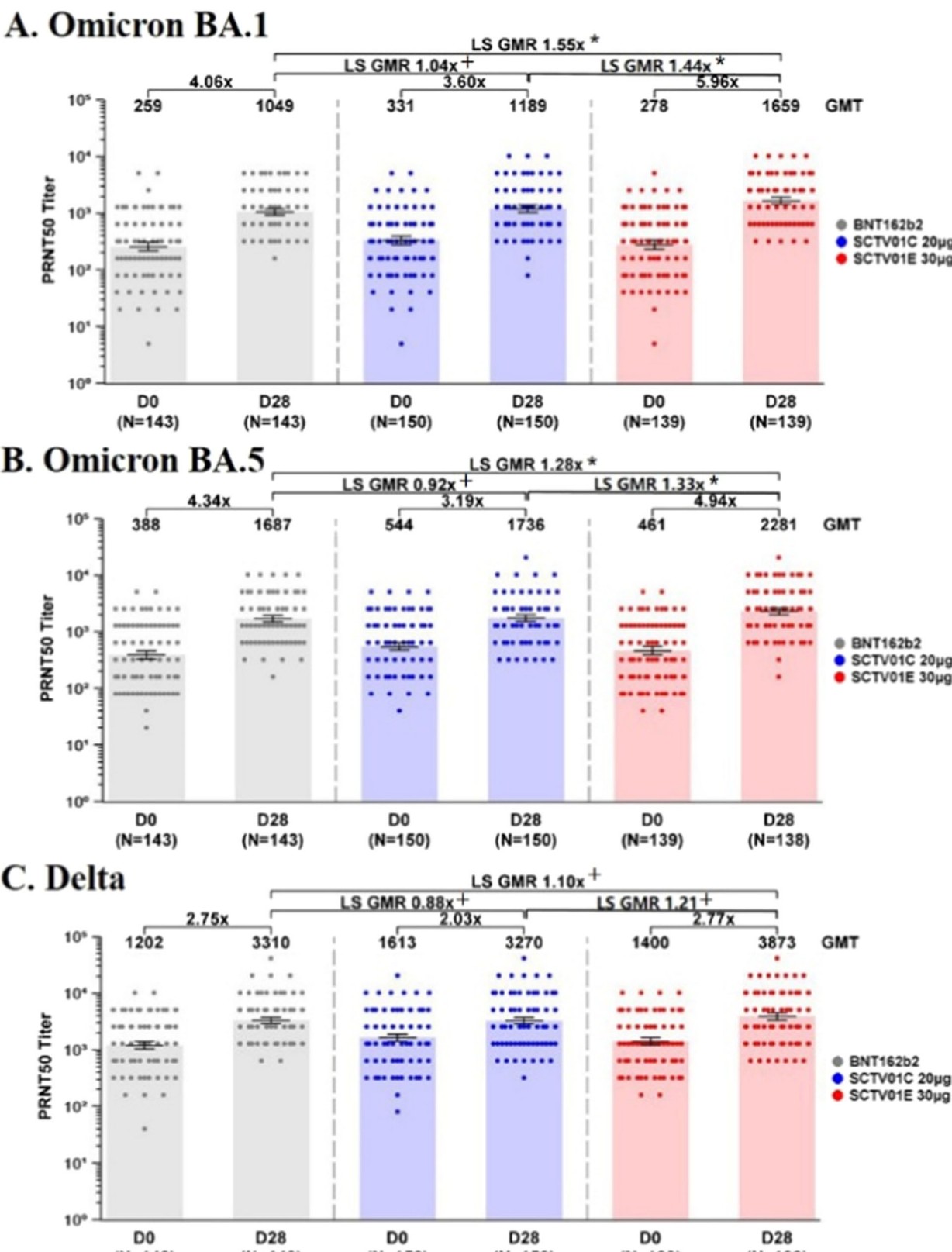

**Fig. 2 | GMTs of live virus nAb against Omicron BA.1, BA.5, and Delta. A** The geometric mean titers (GMTs) of live virus neutralizing antibodies (nAb) against Omicron BA.1 at day 28 post-injection measured using 50% plaque reduction neutralization test (PRNT50). **B** The GMTs of nAb against Omicron BA.5 at day 28 post-injection. **C** The GMTs of nAb against Delta variant at day 28 post-injection. For (**A**–**C**), bars show the GMTs with 95% CIs at day 0 and day 28. Dots represent the values for individual participants. Centre of the error bars represents the GMT. Only those with available baseline and post-baseline data were included in BNT162B2 group (grey), SCTV01C group (blue) and SCTV01E group (red). GMR geometric mean ratio, LS GMR least square geometric mean ratio. Note: Subjects who were COVID-19 infected between Day 0 and Day 28 were excluded from analysis. Source data are provided as a Source Data file. *superiority; †non-inferiority.

**Table 2 | Seroresponse to Omicron BA.1, BA.5 and Delta**

| n-Seroresponse | BNT162b2 | SCTV01C | SCTV01E |
|---|---|---|---|
| Omicron BA.1 variant | | | |
| Seroresponse rate, n (%) | 88 (61.5) | 78 (52.0) | 106 (76.3) |
| 95% CI[a] | 53.0, 69.5 | 43.7, 60.2 | 68.3, 83.1 |
| Increase over BNT162b2 (95% CI)[b] | | −9.3 (−20.4, 1.8) | 14.7 (4.0, 25.5) |
| P value[b] | | 0.1036 | 0.0076 |
| Omicron BA.5 variant | | | |
| Seroresponse rate, n (%) | 95 (66.4) | 69 (46.0) | 87 (63.0) |
| 95% CI[a] | 58.1, 74.1 | 37.8, 54.3 | 54.4, 71.1 |
| Increase over BNT162b2 (95% CI)[b] | | −19.6 (−30.5, −8.7) | −4.4 (−15.4, 6.6) |
| P value[b] | | 0.0006 | 0.4354 |
| Delta variant | | | |
| Seroresponse rate, n (%) | 65 (45.5) | 52 (34.7) | 68 (48.9) |
| 95% CI[a] | 37.1, 54.0 | 27.1, 42.9 | 40.4, 57.5 |
| Increase over BNT162b2 (95% CI)[b] | | −10.5 (−21.9, 0.8) | 2.6 (−9.0, 14.2) |
| P value[b] | | 0.0679 | 0.6576 |

Seroresponse for participants with pre-dose <LLOQ is defined as equal to or above LLOQ; seroresponse for participants with pre-dose ≥ LLOQ is defined as ≥4-fold in titers compared to pre-dose titer.
[a]95% CI of seroresponse rate is based on Clopper-Pearson exact method.
[b]The comparison is based on Cochran-Mantel-Haenszel test (CMH) stratified by randomization stratification factors.

SCTV01E administered to adults who previously received authorized mRNA vaccines or inactivated vaccines. The interim analysis results of cohort 2 are present. Details on inclusion and exclusion criteria are provided in the protocol (Supplementary Materials). Briefly, eligible participants for cohort 2 were aged 18 years and older adults and previously vaccinated with 1, 2 or 3 doses of mRNA COVID-19 vaccine (Pfizer BNT162b2 or Moderna mRNA-1273) and/or previously diagnosed with COVID-19 3–24 months before. Participants with test positive (real-time polymerase chain-reaction assay) for COVID-19 during screening period, fever within three days, with history of allergic reactions to any vaccine or drug and history of infection or disease related to severe acute respiratory syndrome (SARS), Middle East respiratory syndrome (MERS) and HIV-positive were excluded.

The trial is conducted in accordance with the ethical requirements of Good Clinical Practice and the Declaration of Helsinki. The protocol, informed consent and amendments were approved by the Ministry of Health and Prevention (reference number: RCMOHP/CT1/0123/2021). All participants enrolled voluntarily and provided written informed consent before any study procedure.

### Randomization and masking
Eligible participants were randomized to three groups to receive one dose of BNT162b2 (0.3 mL), 20 μg SCTV01C (0.5 mL), or 30 μg SCTV01E (0.5 mL) by a ratio of 1:1:1 using the Interactive Network Response System (IWRS). The participants were stratified by age (18–54 years, ≥55 years), the number of doses of previously received COVID-19 vaccines (0, 1, 2, or 3), the previous COVID-19 (yes or no) history, the interval between previous vaccination and the study vaccination (3–5 months, 6–8 months, 9–12 months, 13–24 months) and baseline nAb level. The randomization codes were generated using block randomization using SAS software (version 9.4). The syringes used for injection were identical in appearance and covered with stickers for masking the solution insides. All participants, investigators, clinical research associates, data analysts, and laboratory staff were blinded to group assignment.

### Procedures
SCTV01C and SCTV01E are recombinant protein vaccines developed and manufactured by Sinocelltech Ltd. in Chinese hamster ovary (CHO) cells (These cell lines have not been identified as misidentified by the International Cell Line Authentication Committee) according to good manufacturing practice guidelines. The main active ingredients of SCTV01C comprise trimeric spike protein S-ECD of SARS-CoV-2 variants Alpha (B.1.1.7) and Beta (B.1.351). SCTV01E has a tetravalent design containing the S-ECD sequences of the Alpha, Beta, Delta, and Omicron BA.1. Both vaccine candidates are adjuvanted with a squalene-based oil-in-water emulsion SCT-VA02B. SCTV01C and SCTV01E were supplied in single use vials as a sterile, emulsified, white solution, 0.5 mL/vial, stored and transported at 2–8 °C protected from light, with a validity period of 24 months. BNT162b2 was used as positive control and the dosage form, package and route of administration were consistent with those of the study vaccines.

One day before vaccination, all participants received a full physical examination, and provided blood samples for baseline safety laboratory testing. Participants were randomized to three subgroups to receive one dose of BNT162b2, 20 μg SCTV01C, or 30 μg SCTV01E at a ratio of 1:1:1. Post injection, solicited adverse event (AE) within 7 days, unsolicited AEs within 28 days, SAE and AESI within 180 days were monitored and recorded. AEs and abnormal changes in laboratory tests were graded according to the FDA Standard[26]. Serum samples were collected to evaluate the geometric mean titers (GMT) of nAb activities against live SARS-CoV-2 Delta, Omicron BA.1 and BA.5 variants on days 0, 28, and 180 using plaque reduction neutralization test (PRNT). The peripheral blood mononuclear cells of the first 150 participants were collected and Th1 (interferon gamma (IFN-γ) release) and Th2 (interleukin-4 (IL-4) release) responses were measured before and at day 28 post-boost, using T-SPOT.COVID test and enzyme-linked immunospot (ELISpot) IL-4 COVID TEST assay. For the Th1 (IFN-γ release) test, spike antigens were used as stimulation antigens along with bovine serum albumin and antimicrobial agents. For the Th2 (IL-4 release) test, spike protein peptides were used for stimulation. The live virus neutralization and ELISpot assays were performed according to the supplier's guidelines (Biogenix,Abu Dhabi, United Arab Emirates) as previously described[27, 28]. In detail, the PRNT assay was verified and performed by Biogenix Labs and G42 Healthcare. The serum samples were first exposed to a 30-minute incubation at 56 °C in a water bath. The sera were then initially diluted five times and then serially diluted from 1:10 to 1:640. These dilutions were mixed with SARS-CoV-2 variants (Delta, Omicron BA.1, and BA.5) and transferred in duplicate to sub-confluent Vero E6 cell monolayer plates. Following an incubation period of 3 – 5 days at 37 °C °Cand 5% CO2 in 6-well plates, antibody titers were determined as the highest serum dilution that resulted in >50% (PRNT50) reduction in the number of plaques compared to the negative control. The negative control had a plaque count ≥50, while the positive control had a plaque count ≤ 50% of the negative control. A cut-off for positivity was established at 1:20. ELISpot assays were performed in cryopreserved peripheral blood mononuclear cells (PBMCs). The cells were rapidly thawed and rested overnight before being stimulated with a pool of peptides containing Spike antigens (the SARS-CoV-2 ancestral strain), bovine serum albumin, and antimicrobial agents. The cells were then incubated at 37 °C for 24–48 h. Phytohemagglutinin (PHA)-stimulated cells were used as a positive control during the assay. To detect IFN-γ and IL-4, mouse monoclonal antibodies (Oxford Immunotec UK, lot numbers: VEC7000001 and VEC7000003, catalog number: COV.435/300) were used following the manual protocols. The spots secreted by the antigen-specific T cells were counted directly from the well using a stereomicroscope or from a digital image captured from a microscope or plate imager. The analysis included only subjects with both baseline and post-baseline data, and the counting results were reported as spots per million PBMCs.

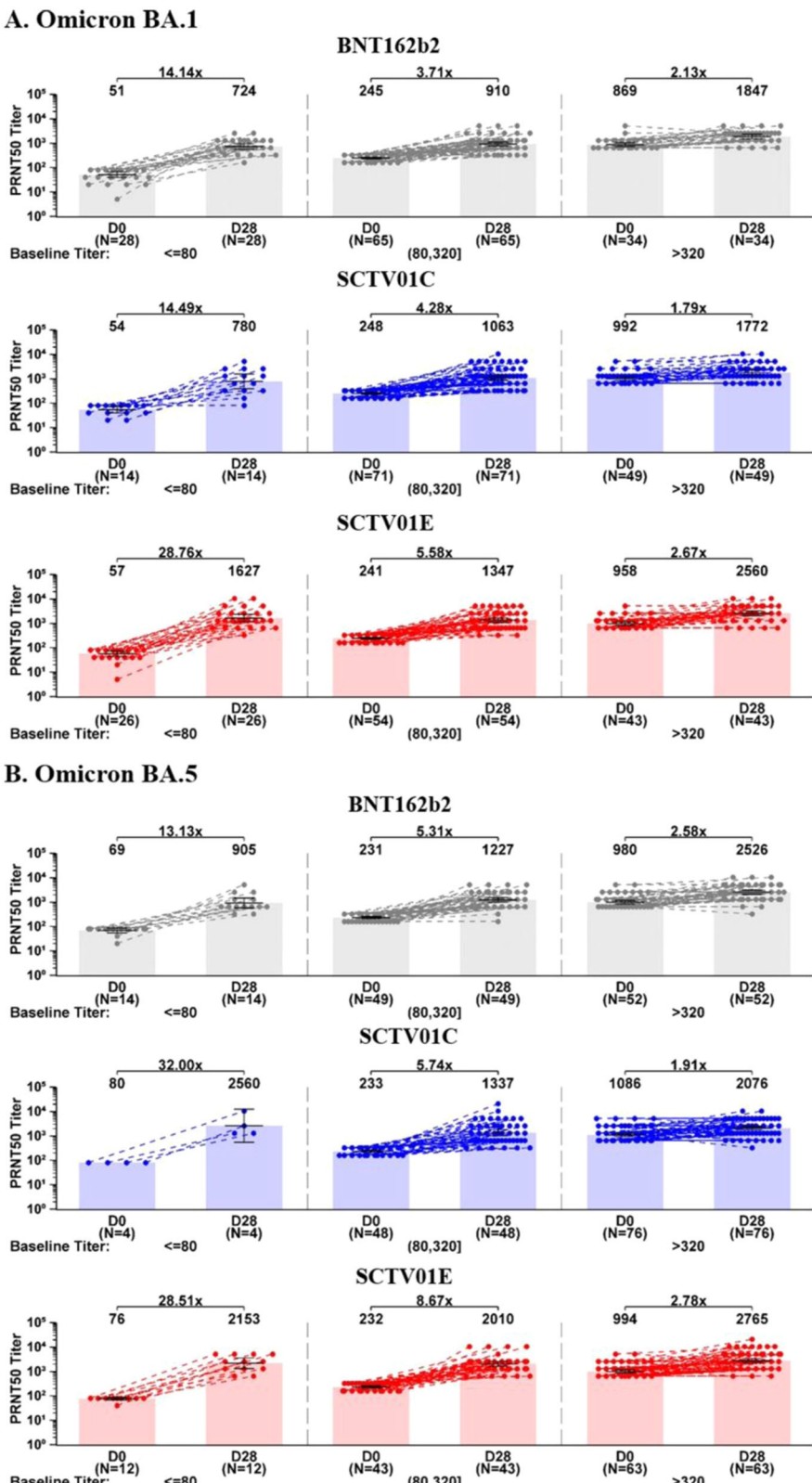

**Fig. 3 | GMTs of neutralizing antibodies against live Omicron BA.1 and BA.5 in groups with low, medium and high baseline titers. A** The geometric mean titers (GMTs) of live virus neutralizing antibodies (nAb) against Omicron BA.1 at day 28 post-injection in groups with low, medium and high baseline titers. **B** The GMTs of nAb against Omicron BA.5 at day 28 post-injection in groups with low, medium and high baseline titers. For (**A, B**), participants from BNT162B2 group (grey), SCTV01C group (blue) and SCTV01E group (red) were assigned to three groups based on their GMT levels at baseline. GMTs at baseline equal to or lower than 4 times of LLOQ (80), in the range of 80–320 and over 320 were considered as low, medium and high baseline titers, respectively. Centre of the error bars represents the GMT, and error bars indicate 95% confidence interval. Note: Subjects who were COVID-19 infected between Day 0 and Day 28 were excluded from analysis. Source data are provided as a Source Data file.

Validation of these assays was conducted at G42 LABORATORY LLC in Abu Dhabi, United Arab Emirate. The validation parameters were in accordance with the EMA/FDA guidance on biomarker assays, which included the estimation of low/maximum limit of detection, precision and accuracy, limits of quantification, dilution linearity, stability, and interference. Optimization was performed for days of cell seeding, working viral dilution, and time for infection. Intra- and inter-assay precision were evaluated and deemed acceptable, as 100% of observed results were within a twofold difference of the tested samples. Positive controls were included with each run to monitor the assay's performance during sample analysis.

### Outcomes

The primary endpoints were the GMT of live virus neutralizing antibodies to Delta (B.1.617.2) and Omicron BA.1 at day 28 post injection. The secondary endpoints for safety were the incidence and severity of adverse reactions (ARs) within 7 days; solicited AE within 7 days; unsolicited AE within 28 days; SAE and AE of AESI within 180 days; laboratory abnormalities related AEs within 14 days, after booster vaccination. The secondary endpoints for immunogenicity included GMTs of neutralizing antibodies (nAb) to Delta and Omicron BA.1 at day 180, GMTs of nAb to BA.5 at day 28 and day 28 T-cell responses and seroresponse rates. An independent data and safety monitoring board (DSMB) reviewed the data.

### Statistical analysis

Statistical analyses were performed with descriptive and pre-specified statistical test methods using SAS software (version 9.4). For the safety analysis, all participants who received vaccines were analyzed based on solicited AEs (local and systemic) within 7 days and unsolicited AEs within 28 days after booster vaccination. The proportion of participants with at least one solicited AE of Grade ≥ 3 was reported for each group. Unsolicited adverse events were coded by MedDRA version 24.1 and tabulated by primary system organ class (SOC) and preferred term (PT) for each group. For the immunogenicity analysis, data reported as below the lower limit of detection were imputed as half of the threshold. For participants with a pre-dose titer below LLOQ, seroresponse is defined as a post-dose titer equal to or above LLOQ; for participants with a pre-dose titer equal to or above LLOQ seroresponse is defined as a post-dose titer at least four-fold the pre-dose titer. GMT and geometric mean fold change over baseline with corresponding 95% confidence intervals (95% CI) were provided at each time point. The 95% CIs were calculated based on the t-distribution of the log-transformed values and then back transformed to the original scale for presentation. The comparisons of the GMTs and geometric mean ratios (GMR) across groups were analyzed using Analysis of Covariance (ANCOVA) based on log-transformed data with covariates of the intervention group, age group, number of dose of COVID-19 vaccine received, interval from last COVID-19 vaccination, and baseline values (in log-transformed scale). The comparisons of seroresponse rate across groups were analyzed based on the pre-defined hypothesis, using ANCOVA with covariates of the intervention group, age group, number of dose of COVID-19 vaccine received, interval from last COVID-19 vaccination, and baseline values. The statistical design was based on a superiority design intended to demonstrate that SCTV01E was superior to BNT162b2 and SCTV01C in terms of the GMTs of neutralizing antibody against Omicron BA.1 and BA.5. Post-hoc analysis focused on the D28 nAb responses to Omicron BA.1 and BA.5 based on participants' pre-dose GMT levels. The sample size was determined based on the following assumptions: the standard deviation of GMTs under log10 transformation was 0.4; the low margin of GMR superiority between SCTV01E and mRNA vaccine was 1 and the non-inferiority margin was 0.67; the dropout rate during study was about 10%, with the one-sided type I error of 0.025 and Power of 80%. The Interim analyses were conducted following the acquisition of safety data within 28 days and immunogenicity data on D28 + 3 for each cohort. The results were analyzed by an unblinded team that was independent of the study operation team. This interim analysis was approved by the DSMB, and the results from cohort 1 have been submitted to another medical journal for review.

### Reporting summary

Further information on research design is available in the Nature Portfolio Reporting Summary linked to this article.

## Data availability

The data for this study are available as a Source Data file and have been deposited in Figshare repositories. The trial has been registered on ClinicalTrials.gov under the identifier NCT05323461. Source data are provided with this paper.

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

## Acknowledgements
This study was funded by the Beijing Science and Technology Planning Project (Z221100007922012) and the National Key Research and Development Program of China (2022YFC0870600). It should be noted that the funding award has been granted to the study itself and not to any individual author or researcher. Sinocelltech Ltd. sponsored this study (NCT 05323461). The sponsor contributed to trial design, data analyses, and data interpretation. We thank the CRO team of PDC FZ-LLC, for their hard work, support, and guidance; Mr. Bo Zhong, the project manager of the trial for his committed dedication to managing and running of the trial; Mr. Adham Rezk and Revonbio B.V. for vaccine consultancy, strategy and coordination. We also acknowledge medical writing and editorial support by Dr. Xiaomei Yang and Dr. Miaomiao Zhang.

## Author contributions
S.H. was the study site principal investigators, responsible for the supervision of the study, the coordination of resources, data analysis, data verification, and interpretation. L.S., A.Ab., A. Al., and A.H. contributed to participant management and implementation of the study (vaccine management, vaccination, participant screening management, communication, and coordination with the sponsor, CRO, and ethics committee). S.A.M. was responsible for laboratory testing and assay development. D.I. and M.A. contributed to the revision and interpretation of data for this work. S.X., J.L., D.L., A.A.H.B., and Y.C. contributed to the medical management, protocol writing, statistical analysis, and manuscript drafting. Q.Z. and L.X. contributed to the study conception, study design, project management, data interpretation, and manuscript writing. All authors critically reviewed and approved the final version of the manuscript.

## Competing interests
L.X. has ownership or potential stock option interests in the company. All other authors declare no other conflicts of interest.
