## [Peer Review File · Nature Communications]

Safety and immunogenicity of a tetravalent and bivalent SARS-CoV-2 protein booster vaccine in menReviewers' Comments:

Reviewer #1:

Remarks to the Author:

The manuscript of Suad Hannawi et al. 'Comparing a Tetravalent 1 SARS-CoV-2 Protein Vaccine with Bivalent Protein Vaccine and Monovalent mRNA Vaccine' describes the immunogenicity and safety profile of two multi-valent COVID-19 vaccines, developed by Sinocelltech, China. The bi-valent vaccine recently received Emergency Use Authorization from the National Health Commission of China as a booster dose for subjects primed with inactivated vaccines.

Developing multi-valent and broader protective COVID-19 vaccines are essential in the current epidemiological situation, as highlighted by the recent Vaccines and Related Biological Products Advisory Committee meeting (January 26, 2023).

The authors demonstrated broad protection caused by the tetravalent COVID-19 vaccine against Omicron BA.1 and BA.5 variants and the significant potential of this vaccine to prevent COVID-19 due to new emergent SARS-CoV-2 strains.

The manuscript is well-written and provides essential information for healthcare providers, public health officials, and vaccine developers.

The manuscript triggers several questions and requests for clarification:

1. The authors may want to elaborate on the selection of vaccine composition (variants Alpha, Beta, Delta, and Omicron BA.1) in the current epidemiological situation in China and globally. It is crucial, considering the recent emergence and broad spread of BQ1.1 and XBB variants with significant antigenic distance from BA.1 and other historical variants

(<https://doi.org/10.1101/2022.11.23.517532>). Do the authors consider the current study as a proof-of-concept for tetravalent recombinant protein vaccine, which requires adaptation to the current epidemiological situation, or do they believe that the proposed vaccine composition is suitable in the current environment?

2. The study manuscript does not include a composition of candidate vaccines. From the paper, it is unclear whether the investigational product includes a combination of recombinant proteins with a squalene-based oil-in-water adjuvant system or whether it is a fusion protein with multiple epitopes. If the vaccine includes a combination of recombinant proteins, the quantity of each protein should be specified.

3. Lines 163-164 'The frequency of solicited AEs were 15 (10.1%) in BNT162b2 group, 19 (12.3%) in SCTV01C group, and 7 (4.8%) in SCTV01E group'. The frequency of solicited AEs after the BNT162b2 vaccine is significantly lower than reported in multiple other studies (e.g., BNT162b2 [COMIRNATY (COVID-19 Vaccine, mRNA)] Evaluation of a Booster Dose (Third Dose), VACCINES AND RELATED BIOLOGICAL PRODUCTS ADVISORY COMMITTEE BRIEFING DOCUMENT, MEETING DATE: 17 September 2021). The authors need to elaborate on potential reasons for the lower reactogenicity observed in the study and the generalizability of the study results.

4. Lines 231-232 The authors refer to the investigational vaccine as thermostable (A thermostable tetravalent protein-based vaccine, SCTV01E, designed to provide...). However, the provided protocol defined storage conditions for the vaccines as 'Stored and transported at 2~8°C away from light'. Authors may need to clarify the thermostability claim (whether the storage conditions for the bivalent vaccine include room temperature and whether any national regulatory agency accepted this claim).

5. Result section: Although statistically significant differences were observed in post-booster antibody titers between study groups, all three tested vaccines induced a robust immune response against Omicron BA.1 and BA.5 variants in the range 1049-1659 for BA.1 and 1687-2281 for BA.5. Inter-group differences between SCTV01E and comparator vaccines do not exceed 1.55 (Figure 3). It is unclear whether these numerically higher titers are expected to be translated into superior clinical efficacy or favorable durability of protection. The author's position on the clinical significance of the observed differences is important and should be presented in the manuscript.

6. Lines 208-219: Authors performed post hoc immunogenicity analysis and stated 'The nAb responses with SCTV01E were consistently superior to those with SCTV01C and BNT162b2, irrespective of baseline GMTs levels of the participants'. This analysis needs some improvement. Firstly, the statement that 'participants were assigned to three groups' needs to be clarified, as subjects were

split into three groups for each tested strain. A low titer group for the BA.1 variant is not necessarily the same as for BA.5 or Delta variants. Authors may consider changing the graphical approach for the presentation of data in Figure 3 and present log-transformed pre-booster titers vs. log-transformed post-booster titers for each tested vaccine to assess whether SCTV01E provides benefit across the whole range of tested titers.

Alternatively, if the authors prefer to keep the current Figure 3, the number of subjects in each selected cohort should be included.

7. Figure 1 - reasons for exclusion from immunogenicity analysis should be added for transparency reasons.

Overall, the manuscript provides important information about the feasibility of developing an adjuvanted tetravalent COVID-19 vaccine as one of the options for long-term COVID-19 control.

Possible example for Figure 3 result presentation (axis X – pre-booster titer; axis Y – post-booster titers). The study size should be sufficient to model response across the whole pre-booster titers.

If parallelism will be demonstrated it will support the conclusion about superior immune response regardless of pre-booster titers.

If the test indicates a lack of parallelism, the estimated effect as a function of the pre-booster titer.

Reviewer #2:

Remarks to the Author:

Overall, a clearly-described and thoughtfully designed clinical study. There are a number of methodological questions that were not sufficiently clearly described to be confident of the soundness of the approach, the validity of the conclusions, nor the possibility of reproducing the results.

The authors describe a portion of a phase III study evaluating the safety and immunogenicity of bivalent and tetravalent recombinant protein COVID-19 variant booster vaccines containing a novel squalene-containing adjuvant compared to a licensed ancestral strain monovalent mRNA booster. By description, the bivalent vaccine has been authorized or approved for use in the Peoples Republic of China. This appears to be the first in human study of the tetravalent booster. The relatively limited data on recombinant protein vaccines, particularly variant vaccines, give this manuscript novelty.

The rationale for the selection of antigen dose of the tetravalent booster is not clearly described in the manuscript or protocol, particularly what data supported the 3-fold higher antigen dose of Omicron BA.1.

The highly skewed participant demographics raise a concern for generalizability. There was no data presented in female nor among older adults. Data among participants with underlying chronic medical conditions are also lacking.

The authors should provide additional detail on the reasons for exclusion from the randomized population, particularly as the principal reason for exclusion could include COVID-19 infection. As the group with the large number of participants excluded was the tetravalent booster, this would be relevant information for readers. In addition, if data are available regarding COVID-19 infections occurring in the available follow up time, it would be relevant to provide these data.

Based on the results provided, there appear to be no specific safety concerns. It would be relevant to address whether there were any findings of concern noted with biological safety monitoring in prior human studies, as well as whether that was considered in the current trial, given the mention of glomerulonephritis in pre-clinical studies.

The primary objectives were based on immunological comparisons. It would be relevant to the reader to describe the validation status of assays used to evaluate immune responses.

Statistical review would be important. It was not really possible for me to follow the sequence of hypothesis testing. Relatedly, it was not clear if the adjusted ANCOVA results presented are the results of the actual hypothesis testing or additional post hoc analysis. As a result, I am not confident in interpreting the p-values presented in the manuscript. It might help to present a table of the results of each of the specified hypothesis tests in the supplementary materials.

Specific comments:

Line 49 – CT.gov indicates mRNA-1273 comparator. Protocol does not specify the comparator. Manuscript states BNT162b2 as the comparator. Please clarify the differences between these sources and explain why the public disclosure is different to the manuscript.

Line 182 – “The pre-181 specified statistical success criteria were met for superiority of GMTs with SCTV01E 182 against Omicron BA.1 and BA.5 compared to those with BNT162b2 and SCTV01C”: unclear what this means. I did not see pre-specified non-inferiority nor superiority hypotheses comparing SCTV01E versus SCTV01C

Line 206 – please clarify what this p value is evaluating. No hypothesis testing was described for seroresponse rate differences.

Line 210 – “assigned to three groups based on their pre-dose GMT levels”: please clarify if this was based on baseline GMT for the respective variant or a single reference strain such as the ancestral strain

Line 224 – please describe the peptides used for stimulation

Line 234 – may be useful to specify that the comparator vaccine is an ancestral strain monovalent mRNA booster

Line 273 – “a large portion of the trial 273 participants may have asymptomatic infection according to published reports.”: what evidence, given baseline PCR ± serology testing, is there of high prevalence of asymptomatic infection?

Line 345 – is the volume injected the same for all groups?

Line 384 – the definition of seroresponse merits some justification. Would it be more uniform if SR denoted a 4-fold rise over the baseline value. If the baseline titer were <LLOQ and thus the baseline value were imputed to ½ LLOQ, would it not require that the D28 titer be ≥2-fold LLOQ, rather than == LLOQ?

Reviewer #3:

Remarks to the Author:

The paper: Comparing a Tetravalent SARS-CoV-2 Protein Vaccine with Bivalent Protein1 Vaccine and Monovalent mRNA Vaccine by Liangzhi Xie et al., represents an interesting study with relevant information concerning the potential benefit of a booster COVID-19 vaccine. The vaccine candidate is tetravalent and use S-protein ectodomain of 4 different VOC, including original Wuhan, beta, delta and omicron compare with a similar protein with 2 VOC similar protein from alpha and beta, as well as RNA vaccine as a booster of individuals previously vaccinated with a full series of BNA vaccine.

A total of 450 participants were included and I the trial with a vaccine containing the S-ECD of the Alpha, Beta, Delta and Omicron BA.1.127 adjuvanted with a squalene-based oil-in-water emulsion and possess a trimerization auxiliary domain (T4-Foldon). The paper is well written and the information provide is relevant to the community working with COVID-19 vaccines.

The protein vaccines can play an important role as a booster and this is an excellent example.

The study shed light on the correspondence between the protein antigen (VOC specific) used and the more efficient neutralization capacity vs the same VOC. Nevertheless, the study also demonstrated that a booster dose with a protein antigen (VOC specific) used, without exact correspondence with the same variant, also increase dramatically but to a lesser extend the specific antibodies and the neutralizing capacity vs it.

In my opinion the authors need also to explain that in order to avoid the misunderstanding of the useless of a booster different with the circulating VOC.

I recoment the paper to be published in the actual form.

Dr Professor Vicente Verez Bencomo

Reviewer #4:

Remarks to the Author:

This trial compares three different vaccines; a tetravalent SARS-CoV-2 protein vaccine, a bivalent protein vaccine, and a monovalent mRNA vaccine. The main goal is to understand how the antibody

response against variants depends on both platform and antigen, in a non-naïve population, which is of interest for SARS-CoV-2 vaccine development. The study was randomized and well powered and demonstrates the advantage of the tetravalent vaccine, in terms of increasing titers to variants, both overall and in subgroup defined by baseline titers. Additional analyses explore the effect of interval and number of prior doses. A strength of this study is the evaluation of the T-cell response.

Minor Comments

Around line 150 you should mention prior vaccination was from mRNA.

In Figure 3 the 1st row should be grey, 2nd row blue, and 3rd row red to be consistent with Figure 2.

Table 2 could have the 1st row be labeled n-seroresponders (%) to avoid confusion. I thought n was number randomized.

I think that an analysis of interest for the phase III clinical endpoints trial will be to compare the durability of the different vaccines.

It would've been nice to have say a bivalent mRNA to disentangle the effects of antigen and platform, but I do understand you have to do what's practical.

Reviewer 1:

Developing multi-valent and broader protective COVID-19 vaccines are essential in the current epidemiological situation, as highlighted by the recent Vaccines and Related Biological Products Advisory Committee meeting (January 26, 2023). The authors demonstrated broad protection caused by the tetravalent COVID-19 vaccine against Omicron BA.1 and BA.5 variants and the significant potential of this vaccine to prevent COVID-19 due to new emergent SARS-CoV-2 strains. The manuscript is well-written and provides essential information for healthcare providers, public health officials, and vaccine developers.

Q1. The authors may want to elaborate on the selection of vaccine composition (variants Alpha, Beta, Delta, and Omicron BA.1) in the current epidemiological situation in China and globally. It is crucial, considering the recent emergence and broad spread of BQ1.1 and XBB variants with significant antigenic distance from BA.1 and other historical variants (<https://doi.org/10.1101/2022.11.23.517532>). Do the authors consider the current study as a proof-of-concept for tetravalent recombinant protein vaccine, which requires adaptation to the current epidemiological situation, or do they believe that the proposed vaccine composition is suitable in the current environment?

Response: The tetravalent vaccine SCTV01E was developed as a modified version of the bivalent (Alpha + Beta) vaccine SCTV01C by adding two subsequent VOC variants Delta and Omicron BA.1. In this clinical study, SCTV01C demonstrated significant cross-neutralizing capability against Omicron BA.1 and BA.5 variants, which emerged two years after the initial development of SCTV01C. The newer version of the vaccine, SCTV01E, showed an even greater breadth of cross-neutralizing capabilities against a variety of Omicron variants in the pre-clinical studies. To date, a total of 7 clinical trials have been conducted with both SCTV01C and SCTV01E, collectively demonstrating their potential as an important vaccine platform in the context of the challenging epidemiological situation where multiple major variants are prevalent simultaneously. This platform's flexibility enables the rapid replacement of up to four new variant antigens to adapt to immune-evading variants. The findings of this investigation suggest that a tetravalent recombinant protein may be an effective approach to address both current and potential future epidemiological challenges. SCTV01E is currently undergoing a phase 3 efficacy study in China. On March 22, 2023, the National Health Commission of the People's Republic of China granted Emergency Use Authorization for SCTV01E, recommending it as a booster dose and as the primary vaccination for individuals who have previously been infected during the COVID-19 pandemic.

In the Discussion section of the revised manuscript, we have included the statement that *“The tetravalent vaccine SCTV01E was developed as a modified version of the bivalent (Alpha + Beta) vaccine SCTV01C by adding two subsequent VOC variants Delta and Omicron BA.1. During this clinical study, SCTV01C demonstrated significant cross-neutralizing capability against Omicron BA.1 and BA.5 variants which emerged two years after its initial development. SCTV01E showed even greater breadth of cross-neutralizing capabilities against a variety of Omicron variants during pre-clinical*

studies. Seven clinical trials have been conducted for both SCTV01C and/or SCTV01E, collectively demonstrating their potential as an important vaccine platform in the context of the challenging epidemiological situation where multiple major variants are prevalent simultaneously. The flexibility of this platform enables rapid replacement of up to four new variant antigens to adapt to immune-evading variants. The findings of this investigation suggest that a tetravalent recombinant protein may be an effective approach to address both current and potential future epidemiological challenges. Currently, a phase 3 efficacy study with SCTV01E is underway in China (NCT05308576).”

Q2. The study manuscript does not include a composition of candidate vaccines. From the paper, it is unclear whether the investigational product includes a combination of recombinant proteins with a squalene-based oil-in-water adjuvant system or whether it is a fusion protein with multiple epitopes. If the vaccine includes a combination of recombinant proteins, the quantity of each protein should be specified.

Response: As per your suggestion, in the Introduction section of the revised manuscript, we have included details about SCTV01E: “*SCTV01E contains a blend of Spike-ECD proteins derived from SARS-CoV-2 variants, Alpha (B.1.1.7), Beta (B.1.351), Delta (B.1.617.2), and Omicron BA.1., in a proportion of 1:1:1:3, with a total quantity of 30 micrograms. The selection of a 1:1:1:3 antigen ratio was based on empirical animal data indicating that a higher dose of Omicron BA.1 antigen is required to elicit an optimal immune response as a booster vaccine against the newer BA.1 variant. Additionally, it is formulated with SCT-VA02B, an adjuvant system consisting of a squalene-based oil-in-water emulsion.*”

Q3. Lines 163-164 'the frequency of solicited AEs were 15 (10.1%) in BNT162b2 group, 19 (12.3%) in SCTV01C group, and 7 (4.8%) in SCTV01E group'. The frequency of solicited AEs after the BNT162b2 vaccine is significantly lower than reported in multiple other studies (e.g., BNT162b2 [COMIRNATY (COVID-19 Vaccine, mRNA)] Evaluation of a Booster Dose (Third Dose), VACCINES AND RELATED BIOLOGICAL PRODUCTS ADVISORY COMMITTEE BRIEFING DOCUMENT, MEETING DATE: 17 September 2021). The authors need to elaborate on potential reasons for the lower reactogenicity observed in the study and the generalizability of the study results.

Response: As per your request, we have incorporated your suggestion into the Discussion section of the revised manuscript. The added text is as follows: "*In this study, 16.8% of participants in the BNT162b2 group reported experiencing at least one treatment-emergent AE, and 19 individuals (12.8%) experienced at least one treatment-related AE. the total frequency of AEs in this study is comparable to that reported in a phase 3 trial of BNT162b2 booster, which found that among 5050 participants, 25.0% experienced at least one AE after receiving a third dose of the vaccine, with 23.4% being related to vaccine administration¹⁹. However, these incidence rates are much lower than those reported in the Vaccines and Related Biological Products Advisory Committee Briefing Document (17 September 2021), which demonstrated that, within one month following the administration of 3rd dose of BNT162b2 vaccine, 77.2% of 306*

participants reported any systemic reaction²⁰. Possible reasons for these inconsistencies could include differences in the definition, measurement, and reporting of AEs across different studies, as well as variations in population characteristics like age distribution, comorbidities, prior vaccination, and infection history. It is worth noting that the high rate of prior infections and the predominance of young male participants in this trial might have contributed to the lower occurrence of AEs observed.”

Reference:

19. Moreira ED Jr, Kitchin N, Xu X, et al. Safety and Efficacy of a Third Dose of BNT162b2 Covid-19 Vaccine. *N Engl J Med*. 2022 May 19;386 (20).

20. FDA. Vaccines and Related Biological Products Advisory Committee September 17, 2021 Meeting Briefing Document. Available online: <https://www.fda.gov/media/152176/download>. (accessed on 5 May 2023)

Q4. Lines 231-232 the authors refer to the investigational vaccine as thermostable (A thermostable tetravalent protein-based vaccine, SCTV01E, designed to provide...). However, the provided protocol defined storage conditions for the vaccines as 'Stored and transported at 2~8°C away from light'. Authors may need to clarify the thermostability claim (whether the storage conditions for the bivalent vaccine include room temperature and whether any national regulatory agency accepted this claim).

Response: Thank you for bringing up this concern. We would like to clarify that the storage conditions for the vaccines in our study were "2~8°C away from light," as defined in the protocol. While our extensive thermostability studies have shown that the vaccine remains stable at room temperature (25°C) for over 6 months, we have not requested regulatory approval for room temperature storage of this vaccine due to the complex nature of such an environment.

We acknowledge that the use of the term "thermostable" may have been misleading in this context. As a result, we have removed this term from our manuscript and any other relevant materials.

Q5. Result section: Although statistically significant differences were observed in post-booster antibody titers between study groups, all three tested vaccines induced a robust immune response against Omicron BA.1 and BA.5 variants in the range 1049-1659 for BA.1 and 1687-2281 for BA.5. Inter-group differences between SCTV01E and comparator vaccines do not exceed 1.55 (Figure 3). It is unclear whether these numerically higher titers are expected to be translated into superior clinical efficacy or favorable durability of protection. The author's position on the clinical significance of the observed differences is important and should be presented in the manuscript.

Response: As per your request, we have included a statement in the Discussion section of the revised manuscript, indicating “*While statistically significant differences in post-booster antibody titers were observed between study groups, further clinical evidence is needed to demonstrate whether the numerically higher antibody titers would lead to superiority in clinical efficacy or durability of protection.*”

Q6. Lines 208-219: Authors performed post hoc immunogenicity analysis and stated 'the nAb responses with SCTV01E were consistently superior to those with SCTV01C and BNT162b2, irrespective of baseline GMTs levels of the participants'. This analysis needs some improvement. Firstly, the statement that 'participants were assigned to three groups' needs to be clarified, as subjects were split into three groups for each tested strain. A low titer group for the BA.1 variant is not necessarily the same as for BA.5 or Delta variants. Authors may consider changing the graphical approach for the presentation of data in Figure 3 and present log-transformed pre-booster titers vs. log-transformed post-booster titers for each tested vaccine to assess whether SCTV01E provides benefit across the whole range of tested titers.

Alternatively, if the authors prefer to keep the current Figure 3, the number of subjects in each selected cohort should be included.

Response: Thank you for bringing up this concern. As per your suggestion, in the revised manuscript, we have updated Figure 3 to include the number of subjects in each selected cohort.

A. Omicron BA.1

B. Omicron BA.5

Fig. 3 GMTs of neutralizing antibodies against live Omicron BA.1 (A) and BA.5 (B) in groups with low, medium and high baseline titers. Participants were assigned to three groups based on their GMT levels at baseline. GMTs at baseline equal to or lower than 4 times of LLOQ (80), in the range of 80-160 and over 160 were considered as low, medium and high baseline titers, respectively.

Q7. Figure 1 - reasons for exclusion from immunogenicity analysis should be added for transparency reasons.

Response: Upon your request, we have revised Figure 1 and included the following details: "A total of 451 participants were randomized into the study, with 149, 154, and 147 participants assigned to receive a single dose of BNT162b2, 20 µg SCTV01C, and 30 µg SCTV01E, respectively (One participant withdrew before vaccination). Notably, six participants in the BNT162b2 group, four in the SCTV01C group, and eight in the SCTV01E group were excluded from the immunogenicity analysis due to missed or out-of-window scheduled visits."

Fig.1 Flow diagram of the participants. Nab, neutralizing antibody.

Reviewer 2:

Overall, a clearly-described and thoughtfully designed clinical study. There are a number of methodological questions that were not sufficiently clearly described to be confident of the soundness of the approach, the validity of the conclusions, nor the possibility of reproducing the results.

The authors describe a portion of a phase III study evaluating the safety and immunogenicity of bivalent and tetravalent recombinant protein COVID-19 variant booster vaccines containing a novel squalene-containing adjuvant compared to a

licensed ancestral strain monovalent mRNA booster. By description, the bivalent vaccine has been authorized or approved for use in the People’s Republic of China. This appears to be the first in human study of the tetravalent booster. The relatively limited data on recombinant protein vaccines, particularly variant vaccines, give this manuscript novelty.

Q1. The rationale for the selection of antigen dose of the tetravalent booster is not clearly described in the manuscript or protocol, particularly what data supported the 3-fold higher antigen dose of Omicron BA.1.

Response: As per your suggestion, in the Introduction section of the revised manuscript, we have included details about SCTV01E: *“SCTV01E contains a blend of spike protein extracellular domains (S-ECD) derived from SARS-CoV-2 variants, Alpha, Beta, Delta, and Omicron BA.1., in a proportion of 1:1:1:3, with a total quantity of 30 micrograms. The antigen composition was designed with consideration of the greater number of mutations found in key neutralizing antibody epitopes in Omicron BA.1, as well as its stronger immune evasion capabilities compared to other variants in existing vaccines. Animal studies have demonstrated that a higher booster dose of Omicron BA.1 antigen is necessary to elicit an optimal immune response as a booster vaccine against this newer BA.1 variant.”*

Q2. The highly skewed participant demographics raise a concern for generalizability. There was no data presented in female nor among older adults.

Response: The population of this clinical trial in the UAE consisted predominantly of males, possibly due to cultural or societal factors limiting the participation of women in this clinical trial. Additionally, the trial was conducted in regions with a high concentration of migrant workers who might be more willing to participate in the COVID-19 vaccine clinical trials.

The limitations section of the revised manuscript addresses the issue that, *“the study's sample population was mostly composed of young male adults. This lack of diversity may affect the generalizability and applicability of the study results. Although previous clinical studies involving SCTV01C did not reveal any significant differences in AEs or immunogenicity between male and female participants or between younger and older adults, further investigations on SCTV01E with a more balanced demographic representation are necessary.”*

Q3. Data among participants with underlying chronic medical conditions are lacking.

Response: In the revised manuscript, Supplementary Table 2 included information on the participants' underlying health conditions. The Results section notes that *“Out of 451 participants, only four had chronic medical conditions (diabetes), with three in the BNT162b2 group and one in the SCTV01C group.”*

Supplementary Table 2. Participants' underlying health conditions

	BNT162b2	SCTV01C	SCTV01E	Overall
System Organ Class	(N=149)	(N=154)	(N=148)	(N=451)
Preferred Term	n (%)	n (%)	n (%)	n (%)

Number of Subjects with Any				
Medical History	8 (5.4)	8 (5.2)	8 (5.4)	24 (5.3)
Surgical and medical procedures	5 (3.4)	3 (1.9)	6 (4.1)	14 (3.1)
Appendicectomy	2 (1.3)	0	1 (0.7)	3 (0.7)
Hernia repair	2 (1.3)	0	1 (0.7)	3 (0.7)
Inguinal hernia repair	0	0	2 (1.4)	2 (0.4)
Spinal operation	0	0	2 (1.4)	2 (0.4)
Intervertebral disc operation	0	1 (0.6)	0	1 (0.2)
Knee operation	0	1 (0.6)	0	1 (0.2)
Limb operation	0	1 (0.6)	0	1 (0.2)
Surgery	1 (0.7)	0	0	1 (0.2)
Immune system disorders	1 (0.7)	1 (0.6)	2 (1.4)	4 (0.9)
Food allergy	0	1 (0.6)	2 (1.4)	3 (0.7)
Dust allergy	1 (0.7)	0	0	1 (0.2)
Metabolism and nutrition disorders	1 (0.7)	3 (1.9)	0	4 (0.9)
Diabetes mellitus	1 (0.7)	3 (1.9)	0	4 (0.9)
Gastrointestinal disorders	0	1 (0.6)	1 (0.7)	2 (0.4)
Intestinal obstruction	0	1 (0.6)	1 (0.7)	2 (0.4)
Pregnancy, puerperium and perinatal conditions	0	1 (0.6)	0	1 (0.2)
Previous caesarean section	0	1 (0.6)	0	1 (0.2)
Skin and subcutaneous tissue disorders	1 (0.7)	0	0	1 (0.2)
Vitiligo	1 (0.7)	0	0	1 (0.2)

Q4. The authors should provide additional detail on the reasons for exclusion from the randomized population, particularly as the principal reason for exclusion could include COVID-19 infection. As the group with the large number of participants excluded was the tetravalent booster, this would be relevant information for readers.

Response: Upon your request, in the Methods section, we have revised Figure 1 and added information regarding the reasons for exclusion from the randomized population: *"A total of 451 participants were randomized into the study, with 149, 154, and 147 participants assigned to receive a single dose of BNT162b2, 20 µg SCTV01C, and 30 µg SCTV01E, respectively (One participant withdrew before vaccination). Notably, six participants in the BNT162b2 group, four in the SCTV01C group, and eight in the SCTV01E group were excluded from the immunogenicity analysis due to missed or out-of-window scheduled visits"*.

Q5. If data are available regarding COVID-19 infections occurring in the available follow up time, it would be relevant to provide these data.

Response: During the study, all participants were consistently monitored for symptomatic SARS-CoV-2 infection to ensure timely diagnosis and treatment following the diagnosis and treatment guidelines of the Food and Drug Administration (FDA). However, no cases of COVID-19 infection were reported during the available follow-up period when the data were locked for interim analysis. We have included this information in the Results section of the updated manuscript.

Q6. Based on the results provided, there appear to be no specific safety concerns. It would be relevant to address whether there were any findings of concern noted with biological safety monitoring in prior human studies, as well as whether that was considered in the current trial, given the mention of glomerulonephritis in pre-clinical studies.

Response: Based on your suggestion, we have updated the Discussion section of the manuscript to incorporate the pre-clinical safety data and other clinical study results. *“In our study, we found that all AEs related to SCTV01E were mild or moderate, and no serious AEs were reported. These findings are consistent with previous clinical studies of SCTV01C¹³⁻¹⁵, which identified no new safety concerns. It is important to note that during the repeat-dose toxicity test of SCTV01E in rats, certain abnormalities were observed. These included increases in eutrophilic and eosinophils, fibrinogen, and globulin, as well as decreases in reticulocyte and albumin levels. Additionally, glomerulonephritis was observed in the kidneys of two out of twenty rats, however, these changes were not observed in the present trial.”*

13. Wang G, Zhao K, Han J, et al. Safety and immunogenicity of a bivalent SARS-CoV-2 recombinant protein vaccine, SCTV01C in unvaccinated adults: A randomized, double-blinded, placebo-controlled, phase I clinical trial [published online ahead of print, 2022 Nov 17]. *J Infect.* 2022;S0163-4453(22)00649-1.

14. Hannawi S, Saifeldin L, Abuquta A, et al. Safety and immunogenicity of a bivalent SARS-CoV-2 protein booster vaccine, SCTV01C, in adults previously vaccinated with mRNA vaccine: A randomized, double-blind, placebo-controlled phase 1/2 clinical trial. *EBioMedicine* 2022;87: 104386.

15. Hannawi S, Saifeldin L, Abuquta A, et al. Safety and immunogenicity of a bivalent SARS-CoV-2 protein booster vaccine, SCTV01C in adults previously vaccinated with inactivated vaccine: A randomized, double-blind, placebo-controlled phase 1/2 clinical trial. *J Infect* 2022;S0163-4453(22)00693-4.

Q7. The primary objectives were based on immunological comparisons. It would be relevant to the reader to describe the validation status of assays used to evaluate immune responses.

Response: As per your suggestion, we have included additional details regarding the validation of assays in the Methods section of the revised manuscript: *“The live virus neutralization assay and ELISpot assay for Th1/Th2 responses were performed according to the supplier’s guidelines (Biogenix, Abu Dhabi, United Arab Emirates) as*

previously described. Validation of these assays was conducted at G42 LABORATORY LLC in Abu Dhabi, United Arab Emirates. The assays underwent validation at G42 LABORATORY LLC located in Abu Dhabi, United Arab Emirates. The validation parameters were in accordance with the EMA/FDA guidance on biomarker assays, which included the estimation of low/maximum limit of detection, precision and accuracy, limits of quantification, dilution linearity, stability, and interference. Optimization was performed for days of cell seeding, working viral dilution, and time for infection. Intra- and inter-assay precision were evaluated and deemed acceptable, as 100% of observed results were within a 2-fold difference of the tested samples. Positive controls were included with each run to monitor the assay's performance during sample analysis.”

Q8. Statistical review would be important. It was not really possible for me to follow the sequence of hypothesis testing. Relatedly, it was not clear if the adjusted ANCOVA results presented are the results of the actual hypothesis testing or additional post hoc analysis. As a result, I am not confident in interpreting the p-values presented in the manuscript. It might help to present a table of the results of each of the specified hypothesis tests in the supplementary materials.

Response: Following your recommendation, a supplemental table has been included in the supplementary materials of the revised manuscript that summarizes the results of each hypothesis test (Supplemental Table 4).

Supplementary Table 4. Summary of the statistical analysis of the immunogenicity

	Goemetric mean titer ratio GMR (95% CI), P value		
	BA.1	BA.5	Delta
SCTV01E vs. BNT162b2	1.55 (1.30,1.85), P <.0001	1.28 (1.07,1.54), P = 0.0069	1.09 (0.92,1.30), P = 0.3154
SCTV01C vs. BNT162b2	1.04 (0.87,1.24), P = 0.6647	0.92 (0.77,1.10), P = 0.3415	0.88 (0.74,1.05), P = 0.1529

Note: the comparison was performed based on log-transformed titers by employing ANCOVA model with intervention group, randomization stratification factors, and log-transformed baseline titer value as covariates. The GMR and its 95% CI was obtained by anti-log transforming the LS mean difference from ANCOVA model.

Comparison in gray cells met the non-inferiority criteria of 0.67, and others met the superiority criteria of 1.

Specific comments:

Line 49 – CT.gov indicates mRNA-1273 comparator. Protocol does not specify the comparator. Manuscript states BNT162b2 as the comparator. Please clarify the differences between these sources and explain why the public disclosure is different to the manuscript.

Response: Thank you for pointing this out. We apologize for the oversight in not updating the registered information on CT.gov. We would like to inform you that we have switched the comparator vaccine from mRNA-1273 to BNT162b2 due to challenges in obtaining mRNA-1273 vaccine supply and low vaccination rates of mRNA-1273 in the trial regions, which made it difficult to recruit mRNA-1273 vaccinated participants. We have recently updated this information on ClinicalTrials.gov to accurately reflect the change.

Line 182 – “The pre- specified statistical success criteria were met for superiority of GMTs with SCTV01E against Omicron BA.1 and BA.5 compared to those with BNT162b2 and SCTV01C”: unclear what this means. I did not see pre-specified non-inferiority nor superiority hypotheses comparing SCTV01E versus SCTV01C

Response: Thank you for bringing this to our attention. We would like to clarify that the primary objectives of the study were to compare the immunogenicity of SCTV01E and SCTV01C versus BNT162b2. The analysis of SCTV01E versus SCTV01C was performed post-hoc using the same statistical criteria. In the revised manuscript, Line 182 has been updated as follows: “*The pre- specified statistical success criteria were met for the superiority of GMTs with SCTV01E against Omicron BA.1 and BA.5 compared to those with BNT162b2. Additionally, the post-hoc analysis indicated that the GMTs against Omicron BA.1 and BA.5 with SCTV01E were significantly higher than those with SCTV01C.*”

Line 206 – please clarify what this p value is evaluating. No hypothesis testing was described for seroresponse rate differences.

Response: As per your request, we have made the following addition to the Methods section of the revised manuscript: “*The comparisons of seroresponse rate across groups were analyzed based on the pre-defined hypothesis, using ANCOVA with covariates of the intervention group, age group, number of dose of COVID-19 vaccine received, interval from last COVID-19 vaccination, and baseline values.*”

Line 210 – “assigned to three groups based on their pre-dose GMT levels”: please clarify if this was based on baseline GMT for the respective variant or a single reference strain such as the ancestral strain.

Response: The pre-dose GMT levels were based on baseline GMT for the respective variant.

We have included this information in the Results section, indicating, “*Participants were divided into three groups based on their pre-dose GMT levels for each specific variant.*”

Line 224 – please describe the peptides used for stimulation.

Response: We have added the following information to the Methods section: “*For the Th1 (IFN- γ release) test, spike antigens were used as stimulation antigens along with bovine serum albumin and antimicrobial agents. For the Th2 (IL-4 release) test, spike protein peptides were used for stimulation.*”

Line 234 – may be useful to specify that the comparator vaccine is an ancestral strain monovalent mRNA booster

Response: Thank you for your suggestion. We have revised the manuscript to specify that the comparator vaccine used in our study is “an ancestral strain monovalent mRNA booster”.

Line 273 – “a large portion of the trial 273 participants may have asymptomatic infection according to published reports.” What evidence, given baseline PCR \pm serology testing, is there of high prevalence of asymptomatic infection?

Response: During the clinical trial, the Omicron variant had become the dominant strain and replaced the previously prevalent strain. According to studies, asymptomatic individuals caused approximately 32.4% of Omicron infections and 40-45% of SARS-CoV-2 infections (sources cited below). However, there is currently no standard serological test available to differentiate between past vaccination and asymptomatic infection, while PCR sequencing can only detect recent infections. Our study revealed a wide range of baseline neutralizing antibody titers against the Omicron variant at baseline. The GMT levels were notably higher than those reported in earlier studies investigating individuals vaccinated with two doses of mRNA vaccines. Consequently, it is possible that a substantial proportion of trial participants had asymptomatic infections.

In the Limitation section of the revised manuscript, we have updated the text as follows: *“The trial was conducted in an environment of high Omicron variant circulation, and a large portion of the trial participants might have asymptomatic infection according to published reports^{24,25}. Our study revealed a wide range of baseline neutralizing antibody titers against the Omicron variant. Notably, the GMT levels were considerably higher than those reported in earlier studies investigating individuals vaccinated with two doses of mRNA vaccines. However, there was no standard way to differentiate asymptomatic individuals.”*

24. Shang W, Kang L, Cao G, et al. Percentage of Asymptomatic Infections among SARS-CoV-2 Omicron Variant-Positive Individuals: A Systematic Review and Meta-Analysis. *Vaccines (Basel)*. 2022 Jun 30;10(7):1049.

25. Oran DP, Topol EJ. Prevalence of asymptomatic SARS-CoV-2 infection. A narrative review. *Ann Intern Med*. 2020;173:362-7.

Line 345 – is the volume injected the same for all groups?

Response: We have added the following information to the Methods section of the revised manuscript: *“Participants were randomized to three groups to receive one dose of BNT162b2 (0.3 mL), 20 µg SCTV01C (0.5 mL), or 30 µg SCTV01E (0.5 mL) by a ratio of 1:1:1.”*

Line 384 – the definition of seroresponse merits some justification. Would it be more uniform if SR denoted a 4-fold rise over the baseline value. If the baseline titer were <LLOQ and thus the baseline value were imputed to ½ LLOQ, would it not require that the D28 titer be ≥2-fold LLOQ, rather than == LLOQ?

Response: The common practice in determining seroresponse is to use half the LLOQ as a default value when the baseline titer falls below LLOQ. However, we agree that this approach can be confusing when defining seroresponse as a 4-fold increase over LLOQ when the baseline level is below LLOQ. To address this issue in our study, we defined seroresponse as a titer equal to or above LLOQ when the pre-dose titer is below LLOQ. For participants with a pre-dose titer equal to or greater than LLOQ, seroresponse was defined as a four-fold increase in titers post-dose compared to the pre-dose titer. Nonetheless, this issue did not cause significant concern in our study since 99% of participants had positive PRNT50 results at baseline (above LLOQ).

In the revised manuscript, we have included the details of the definition of seroresponse in the Methods section as follows: "*For participants with a pre-dose titer below LLOQ, seroresponse is defined as a post-dose titer equal to or above LLOQ; for participants with a pre-dose titer equal to or above LLOQ, seroresponse is defined as a post-dose titer at least four-fold the pre-dose titer.*"

Reviewer 3:

The paper is well written and the information provide is relevant to the community working with COVID-19 vaccines. The protein vaccines can play an important role as a booster and this is an excellent example. The study shed light on the correspondence between the protein antigen (VOC specific) used and the more efficient neutralization capacity vs the same VOC. Nerveless, the study also demonstrated that a booster dose with a protein antigen (VOC specific) used, without exact correspondence with the same variant, also increase dramatically but to a lesser extend the specific antibodies and the neutralizing capacity vs it. I recommend the paper to be published in the actual form.

Q. The authors need also to explain that in order to avoid the misunderstanding of the useless of a booster different with the circulating VOC.

Response: Thank you for your feedback. We agree that it is important to clarify the potential benefits of a booster vaccine even if it does not directly match the circulating variant of concern (VOC).

In the Discussion section of the revised manuscript, we have included the statement that "*The tetravalent vaccine SCTV01E was developed as a modified version of the bivalent (Alpha + Beta) vaccine SCTV01C by adding two subsequent VOC variants Delta and Omicron BA.1. During this clinical study, SCTV01C demonstrated significant cross-neutralizing capability against Omicron BA.1 and BA.5 variants, which emerged two years after its initial development. SCTV01E showed even greater breadth of cross-neutralizing capabilities against a variety of Omicron variants during pre-clinical studies¹⁸. Seven clinical trials have been conducted for both SCTV01C and/or SCTV01E, collectively demonstrating their potential as an important vaccine platform in the context of the challenging epidemiological situation where multiple major variants are prevalent simultaneously. The flexibility of this platform enables rapid replacement of up to four new variant antigens to adapt to immune-evading variants. The findings of this investigation suggest that a tetravalent recombinant protein may be an effective approach to address both current and potential future epidemiological challenges. Currently, a phase 3 efficacy study with SCTV01E is underway in China (NCT05308576).*"

18. Wang R, Huang H, Yu C, et al. A spike-trimer protein-based tetravalent COVID-19 vaccine elicits enhanced breadth of neutralization against SARS-CoV-2 Omicron subvariants and other variants [published online ahead of print, 2022 Dec 30]. *Sci China Life Sci.* 2022;1-13.

Reviewer 4:

This trial compares three different vaccines; a tetravalent SARS-CoV-2 protein vaccine, a bivalent protein vaccine, and a monovalent mRNA vaccine. The main goal is to understand how the antibody response against variants depends on both platform and antigen, in a non-naïve population, which is of interest for SARS-CoV-2 vaccine development. The study was randomized and well powered and demonstrates the advantage of the tetravalent vaccine, in terms of increasing titers to variants, both overall and in subgroup defined by baseline titers. Additional analyses explore the effect of interval and number of prior doses. A strength of this study is the evaluation of the T-cell response.

Minor Comments:

1. Around line 150 you should mention prior vaccination was from mRNA.

Response: Thank you for pointing this out. The prior vaccination status has been added in the Result section as follows, “*451 participants who had a prior diagnosis of COVID-19 and/or received BNT162b2 vaccines were enrolled.*”

2. In Figure 3 the 1st row should be grey, 2nd row blue, and the 3rd row red to be consistent with Figure 2.

Response: Thank you for bringing it to our attention. Figure 3 has been updated according to your suggestion, with the first row in grey, the second row in blue, and the third row in red, to maintain consistency with Figure 2.

A. Omicron BA.1

B. Omicron BA.5

Fig. 3 GMTs of neutralizing antibodies against live Omicron BA.1 (A) and BA.5 (B) in groups with low, medium and high baseline titers. Participants were assigned to three groups based on their GMT levels at baseline. GMTs at baseline equal to or lower than 4 times of LLOQ (80), in the range of 80-160 and over 160 were considered as low, medium and high baseline titers, respectively.

3. Table 2 could have the 1st row be labeled n-seroresponders (%) to avoid confusion. I thought n was number randomized.

Response: In the revised manuscript, “*n-Seroreponse*” has been added in the first row of Table 2.

4. I think that an analysis of interest for the phase III clinical endpoints trial will be to compare the durability of the different vaccines.

Response: Thank you for bringing up this concern regarding our study. We appreciate your feedback and would like to clarify that the current manuscript presents an interim analysis of the Phase III trial data, designed to evaluate the immunogenicity of the BNT162b2/SCTV01C/SCTV01E vaccines over a period of 28 days.

Assessing the durability of the immune response is an important endpoint of the trial, and this aspect will be evaluated at 180 days in the final reports. As more data becomes available, we will be able to provide a more comprehensive understanding of the long-term effects of these vaccines. Thank you for your interest in our study.

5. It would've been nice to have say a bivalent mRNA to disentangle the effects of antigen and platform, but I do understand you have to do what's practical.

Response: Thank you for your feedback. We agree that a bivalent mRNA vaccine would be an effective means to disentangle the effects of antigen and platform. Nonetheless, as you pointed out, practical considerations such as cost and feasibility need to be taken into account in the design and implementation of clinical trials. At the onset of this study, there were no bivalent mRNA vaccines available in the market.

Reviewers' Comments:

Reviewer #2:

Remarks to the Author:

The authors have addressed much of the feedback on the prior draft of the manuscript. The current draft is clearer and improved. The only item that wasn't fully addressed was the peptide stimulation used to assess Th1 and Th2 cellular responses. It wasn't clear whether the stimulation peptides were based on the ancestral strain or perhaps include peptides that address variant mutations in the spike sequence, and if so, which variants. This may be proprietary to the assay. I couldn't find it in the package insert for the assay.

Reviewer #2 (Remarks to the Author):

Q1. The authors have addressed much of the feedback on the prior draft of the manuscript. The current draft is clearer and improved. The only item that wasn't fully addressed was the peptide stimulation used to assess Th1 and Th2 cellular responses. It wasn't clear whether the stimulation peptides were based on the ancestral strain or perhaps include peptides that address variant mutations in the spike sequence, and if so, which variants. This may be proprietary to the assay. I couldn't find it in the package insert for the assay.

Response: The stimulation peptides were based on the spike antigen of ancestral strain. We have added this information to the Methods.